# ECo-MoE: Embodiment-Conditioned Mixture of Experts Increases the Evolvability of Robots

**Yibin Wang** [1]  **Muhan Li** [1]  **Zihan Guo** [1]  **Sam Kriegman** [1]

## Abstract

In this paper, we introduce a model of evolution and learning in robots that co-optimizes a distribution of latent design vectors (genotypes) and a mixture of control experts (neural modules), which are gated by the latent coordinates of each decoded design (phenotype). This provides a scalable alternative to co-design algorithms that either train an individual policy for every robot, which is inefficient, or a monolithic universal controller for all robots, which results in overly conservative structures and behaviors. Our approach lies somewhere between these two extremes, preserving ancestral knowledge in a unified yet modular framework in which different body plans activate and deactivate different combinations of learned sensorimotor circuits for goal-directed behavior. This allows one part of the controller to be overhauled to better suit new species of designs as they emerge without disrupting the hard-earned knowledge contained within other expert modules. It also allows pretrained expert policies to be directly plugged into the mixture, which can steer evolution into otherwise unexplored areas of latent space containing desired morphological traits. We refer to this process as "evo by demo" and explore how it may be used to guide freeform evolution toward canonical structures defined by the pretrained model. Videos and code can be found at: eco-moe.github.io.

## 1. Introduction

Evolving populations of simulated robots that learn to succeed in specific conditions can generate new theories about how animals solve similar problems in similar conditions.

The morphological features that emerge can also reveal practical design tricks that were previously missed by human designers. Moreover, evolution is the only known process capable of producing generally intelligent agents, a major goal of modern machine learning. However, because evolution requires a population of distinct designs that are recurrently replaced by redesigned offspring, it is much more computationally expensive than learning to control the behavior of an individual agent with a static design.

To make evolution tractable, the resolution of possible morphological features is typically reduced to the point where they retain little to no relevance to real robots or animals. The overall design is usually restricted to either a 2D tiling of springs (Bhatia et al., 2021; Strgar et al., 2024) or rigid stick figures (Wang et al., 2019; Gupta et al., 2021; Yu et al., 2026). Earlier work considered higher resolution morphology—$10^3$ voxels (Cheney et al., 2013) and $27^3$ voxels (Kriegman et al., 2021b)—but completely removed learning, relying instead on evolution to randomly reposition open-loop actuators throughout each design.

The more general idea of jointly co-optimizing a robot's morphology and controller does not necessarily require evolution (Pathak et al., 2019; Zhao et al., 2020; Yuan et al., 2022; Li et al., 2024; Lu et al., 2025; Yu & Kriegman, 2026) and can occur efficiently within a single agent using gradient information from differentiable simulation (Matthews et al., 2023; Kobayashi et al., 2024). These methods however have yet to generalize beyond exceedingly simple design spaces. Moreover, they may struggle to adequately explore sparse or deceptive search landscapes.

Recently, Li et al. (2025) evolved freeform endoskeletal robots with high definition morphology ($64^3$ voxels), using a voxel-based variational autoencoder (VAE; Brock et al. (2016)) to compress the vast space of possible body plans into a continuous latent genome. The complexity of the decoded designs was offset by co-optimizing a "universal" (morphology agnostic) controller (Gupta et al., 2022) alongside the evolving design population. Instead of training a bespoke control policy for each individual design (Gupta et al., 2021; Strgar et al., 2024), a single policy was shared by every design in the population.

---

[1]Northwestern University, Evanston, IL, USA. Correspondence to: the whole team <xenobot-lab@northwestern.edu>.

*Proceedings of the 43rd International Conference on Machine Learning*, Seoul, South Korea. PMLR 306, 2026. Copyright 2026 by the author(s).

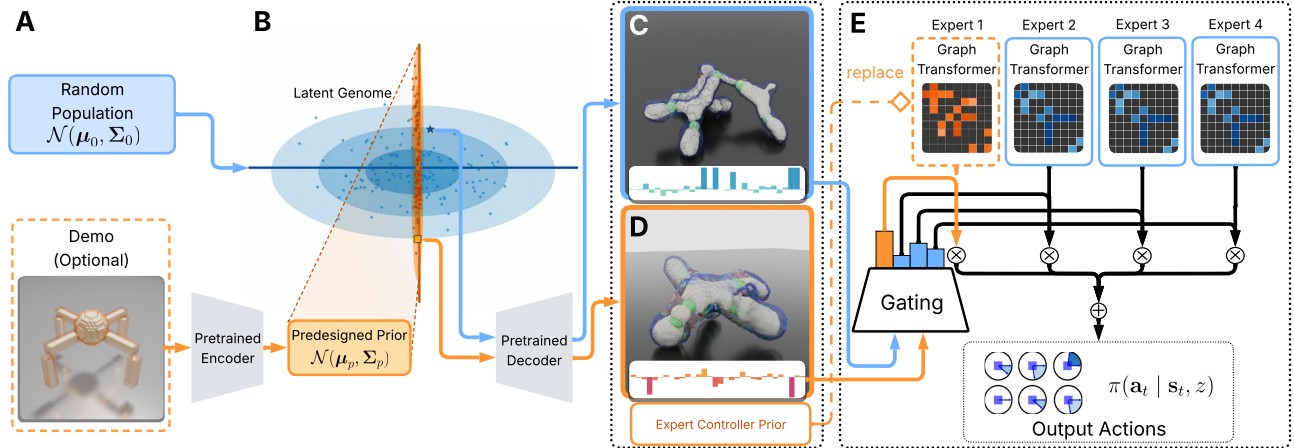

*Figure 1.* **Embodiment-conditioned mixture of experts.** Designs were sampled from an evolving distribution within a latent space of possible genotypes (**A** and **B**). The distribution was initialized randomly for the main experiment (blue region in B); for "evo by demo", it was regularized by a predesigned demo (orange region in B). The latent genotype of each endoskeletal phenotype (**C** and **D**) was fed as input to a gating network that produces a weighted policy output $\pi(a_t \mid s_t, z)$ by mixing expert actions (**E**). In evo-by-demo, a policy was pretrained for the predesigned demo and injected as a frozen expert into the mixture. Designs with latent genes similar to the predefined species were routed with greater weight to the pretrained expert (orange bar in D), steering evolution toward the desired phenotypic traits.

Here, we consider a third, middle approach that replaces the monolithic universal controller used by Li et al. (2025) with a mixture of control experts (Jacobs et al., 1991), modular subnetworks that independently process sensory data and cast a vote for the next action of each motor in the body. Action votes are collected and combined across experts based on the latent genotype of the robot. The main advantage of this approach is that it provides a simple mechanism to dissociate one aspect of control from another, conditioned on the robot's mechanical structure. This modularity allows different parts of the controller to be independently modified, reused, combined and recombined, without destroying the functionality of other parts. This is why, in general, modular systems tend to be more controllable, reconfigurable, robust, adaptive, scalable, and interpretable than nonmodular systems (Suh, 1990; Wagner & Altenberg, 1996; Carroll, 2001; Espinosa-Soto & Wagner, 2010).

By contrast, Xiong et al. (2023) used a hypernetwork to generate the weights of a monolithic base controller (Gupta et al., 2022). Knowledge about how to coordinate distinct kinematic trees was thus entangled across the whole hypernetwork instead of being decomposed into functional modules. Moreover, the robots controlled by Xiong et al. (2023) were predesigned, assumed to be bilaterally symmetrical stick figures, and could not evolve. In related work, Huang et al. (2020) locally controlled each joint in the body across a set of 2D designs using a "shared modular policy", but the same monolithic controller was used at every joint in all body plans. Schaff & Walter (2022) trained a monolithic controller for quadrupeds and hexapods while co-optimizing the length and articulation of their presupposed spine and leg pairs. Wang et al. (2023) optimized a monolithic policy that generated a 2D body and controlled its behavior. Strgar

& Kriegman (2026) pretrained and then finetuned a monolithic universal controller during design evolution on a small, 6-by-6-by-4 voxel grid. Bohlinger et al. (2024) optimized a monolithic policy for a set of commercially available robots with canonical designs.

Because we are primarily interested in the emergence of morphological features that increase fitness, we begin our investigation below by reproducing the results from Li et al. (2025), which represents the state of the art. We then show how modularizing the policy into a mixture of embodiment-conditioned subpolicies can increase the evolvability of robots. Finally, we explore how a pretrained expert can be used to direct evolution toward desired structural and architectural features.

## 2. Methods

In this section, we describe the problem of jointly co-optimizing a universal controller alongside an evolving population of latent designs (Sect. 2.1). We then introduce the embodiment-conditioned mixture of experts (ECo-MoE; Sect. 2.2). Finally, we describe how this framework enables "evo by demo", through which pretrained experts and predesigned bodies can be used to steer evolution (Sect. 2.3).

### 2.1. Joint co-optimization of morphology and control.

We adopt the latent design genome $\mathcal{N}(\mu, \Sigma)$ from Li et al. (2025). Each genotype $z \in \mathbb{R}^{512}$ is decoded as a morphology $m \in \mathcal{M}$, a cartesian tensor of shape $(64, 64, 64, K+2)$, where the last dimension encodes material type (either soft tissue or rigid bone) and $K$ bone IDs, indicating which compound rigid body (i.e. which bone) the rigid voxel belongs

to. After joints are added between opposing bones along the robot's endoskeleton, the robot is placed in a simulated terrestrial environment where its behavior, $\tau$, is driven by the universal controller $\pi_\theta$. In each simulation rollout two objectives are computed: a dense behavior reward $R(\tau)$ used to train the controller (learning) and a sparse design fitness $F(\tau)$ used to update the latent distribution (evolution). Evolutionary update operator $\mathcal{U}_F$ (CMA-ES; Hansen (2016)) and learning update operator $\mathcal{U}_\mathcal{R}^{(n)}$ (PPO; Schulman et al. (2017)) thus co-optimize the expectations,

$$
\begin{aligned}
\mathcal{R}(\mu, \Sigma, \theta) &= \mathbb{E}_z \, \mathbb{E}_{\pi_\theta}\big[R(\tau)\big], \\
\mathcal{F}(\mu, \Sigma, \theta) &= \mathbb{E}_z \, \mathbb{E}_{\pi_\theta}\big[F(\tau)\big].
\end{aligned}
\tag{1}
$$

Evolution operates on a slower timescale (generation $g$) than learning (epoch $e$), with $n$ policy updates per one latent update:

$$
\begin{aligned}
\theta_{e+n} &\leftarrow \mathcal{U}_\mathcal{R}^{(n)}(\mu_g, \Sigma_g, \theta_e), \\
(\mu_{g+1}, \Sigma_{g+1}) &\leftarrow \mathcal{U}_F(\mu_g, \Sigma_g, \theta_{e+n}).
\end{aligned}
\tag{2}
$$

## 2.2. Embodiment-conditioned mixture of experts.

We here introduce an embodiment-conditioned mixture of experts (ECo-MoE) controller, which uses a linear gating layer over $z$,

$$
w_\phi(z) = \mathrm{softmax}(Wz + b), \quad \phi = \{W, b\} \tag{3}
$$

to weight the output of each expert $\pi_\theta^k(a \mid s)$ as follows:

$$
\pi_{\theta,\phi}(a \mid s, z) = \sum_{k=1}^{K} w_\phi^k(z)\, \pi_\theta^k(a \mid s). \tag{4}
$$

Here, each expert is a downsized replica of the graph transformer from Li et al. (2025). We use $K = 4$ experts across all experiments. To prevent expert collapse, we encouraged the gate to produce more distinct routing weights for morphologies with genotypes that are far apart in latent space. For a set of latent genotypes $\{z_g\}$ evaluated in generation $g$, the routing diversity regularized loss was defined as:

$$
\mathcal{L}_{\mathrm{div}} := -\mathrm{mean}\left(\|z_i - z_j\|_2 \cdot \|w_\phi(z_i) - w_\phi(z_j)\|_2\right). \tag{5}
$$

## 2.3. Evo by demo: guiding evolution with a predesigned initialization and a pretrained expert.

Experts can also be pretrained on predefined designs and used to steer evolution as follows. Prior to evolution and learning, we replace the first expert $\pi^0$ in the mixture with a pretrained $\pi_\theta^P$ and freeze its parameters $\theta^P$. We kept the three other experts and the gate trainable. When enabled, the pretrained expert acts as an anchor policy whose usefulness is automatically assessed by the learned gate.

Because the gate is conditioned on the latent genotype ($z$), the scalar gate weight assigned to the pretrained expert, $w_\phi^P(z)$, measures how much the universal controller relies on $\pi_\theta^P$ for a given genotype $z$. We define the optimal compatibility $c^\star(z)$ function under the latent distribution $\mathcal{N}(\mu, \Sigma)$ as the pretrained expert gate weight after optimizing the behavior reward objective:

$$
\begin{aligned}
c^\star(z) &:= w^P\big(z; \phi^\star\big), \\
(\theta^\star, \phi^\star) &\in \arg\max_{\theta,\phi} \mathcal{R}(\mu, \Sigma, \theta, \phi).
\end{aligned}
\tag{6}
$$

In practice we approximate $c^\star(z)$ by using the post-inner-loop gate after $n$ policy updates within generation $g$:

$$
\hat{c}_g(z) := w^P(z; \phi_{g,n}), \tag{7}
$$

where $\phi_{g,n}$ denotes the gate parameters. This provides a learned compatibility score in the range $[0, 1]$, which was used to augment the fitness score as follows:

$$
\tilde{F}(z, \tau) := F(\tau) \cdot \big(\hat{c}_g(z)\big)^\alpha. \tag{8}
$$

We set $\alpha = 2$, which encourages compatibility with the pretrained expert. Thus, if the design is misaligned with $\pi^P$, its genotype will be suppressed during the next evolutionary update (Eq. 2) by the augmented fitness score (Eq. 8).

The predesigned demo used to pretrain an expert can also be encoded into a latent prior (Fig. 6) and used to initialize the mean $\mu_0$ and standard deviation $\Sigma_0$ of the design distribution $\mathcal{N}(\mu, \Sigma)$ with:

$$
\begin{aligned}
\mu_0 &= \begin{cases} \mu^P & \text{when predesigned,} \\ 0 & \text{otherwise,} \end{cases} \\
\Sigma_0 &= \begin{cases} \Sigma^P & \text{when predesigned,} \\ \Sigma & \text{otherwise.} \end{cases}
\end{aligned}
\tag{9}
$$

This provides a structured starting point in latent space even when the predesigned morphology itself is out-of-distribution for the decoder.

# 3. Results

With our experiments, we aim to answer the following three questions. First and foremost, does an ECo-MoE controller affect the evolvability of robots, and if so how? Second, which design choices impact the performance of ECo-MoE? Lastly, to what extent can pretrained experts and predesigned initializations steer evolution toward desired morphological traits?

## 3.1. Performance across task environments.

We begin by comparing the fitness of the designs evolved with an ECo-MoE controller against those evolved with the

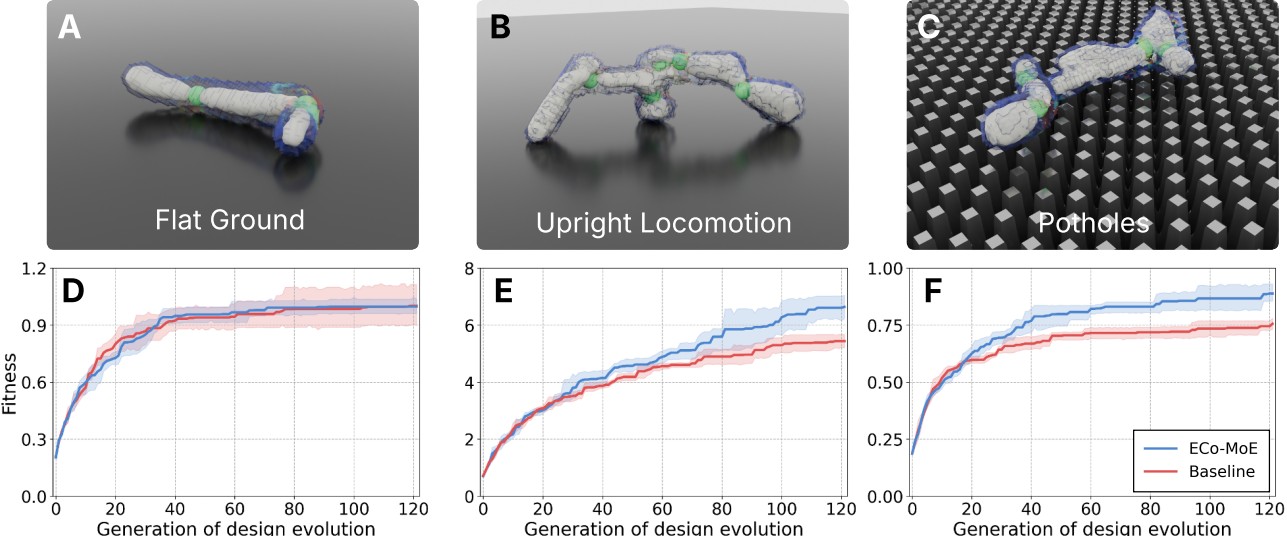

*Figure 2.* **Task environments.** We considered three task environments: Flat Ground (**A**), Upright Locomotion (on flat ground; **B**), and Potholes (**C**). In each one, five independent evolutionary trials were conducted, and the peak fitness achieved by each design was averaged across the population before plotting the cumulative max (higher is better; **D-F**). Evolution with an ECo-MoE controller (blue curves) is compared against evolution with the non-modular (single expert) baseline controller (red); shaded regions indicate 95% bootstrapped confidence intervals. In all three environments net displacement is rewarded. Upright Locomotion adds a second component to the reward function that tracks the proportion of body voxels that fall below a prespecified height threshold during behavior. These results show that, although ECo-MoE fell into the same local optimum as the Baseline on Flat Ground, it significantly increased the evolvability of Upright Locomotion as well as locomotion across Potholes.

state-of-the-art baseline policy from Li et al. (2025). We also use the the task environments from Li et al. (2025): Flat Ground, Upright Locomotion, and Potholes. Flat Ground rewards net displacement on a surface plane. Upright Locomotion increases task difficulty by adding a second reward component that penalizes loss of clearance, as measured by the proportion of body voxels that fall below a prespecified height threshold during behavior. Potholes provides more challenging terrain comprising regularly-spaced depressions ("potholes") along the surface plane.

On Flat Ground, there was not a significant difference between the ECo-MoE and the baseline controller (Fig. 2D). As reported by Li et al. (2025), rewarding solely for displacement on Flat Ground results in the convergent evolution of a homogeneous population of "snakes", which provide a high speed vehicle with low control complexity. In this regime, there is no need for embodiment-conditioned specialization.

In contrast, ECo-MoE significantly increased the evolvability of Upright Locomotion relative to the baseline (Figure 2E). As mentioned above, this reward setting is more difficult than that of Flat Ground because the robot must maintain an upright posture during locomotion; it also generates more diverse designs. These fitness gains suggest that conditioning the controller on embodiment becomes beneficial when sufficient morphology diversity induces heterogeneous control demands. Fig. 3 provides snapshots of the best designs in the population (alongside their fitness, genotype, and expert routing weights) at various points in

evolutionary time.

ECo-MoE also increased the evolvability of locomotion in the Potholes environment (Fig. 2F). This indicates that embodiment-conditioned experts may also become more advantageous as environmental complexity increases.

It is important to note that ECo-MoE does not outperform the baseline simply by scaling model capacity. Across all experiments, its total policy size is comparable to (and slightly smaller than) that of the baseline. Moreover, when considering only the parameters that directly contribute to action generation (i.e. of each expert), the ECo-MoE uses substantially fewer parameters, demonstrating higher parameter efficiency (see Table 1 below). For a fair comparison, we keep the critic architecture and parameter count identical for both methods. See Appx. A for RL training details.

To better understand how ECo-MoE affects evolvability, we visualized how evolution moved through a 2D projection of latent space (obtained by PCA; Fig. 4). Across the five independent initial populations, ECo-MoE and the baseline exhibit distinct evolutionary paths but broadly similar exploration in terms of path length and the area explored by the

*Table 1.* Model size comparison.

| Method | Total policy size | Expert size | No. of Experts | Critic |
|---|---|---|---|---|
| Baseline | 2.563M | 2.563M | 1 | 2.038M |
| ECo-MoE | 2.388M | 0.597M | 4 | 2.038M |

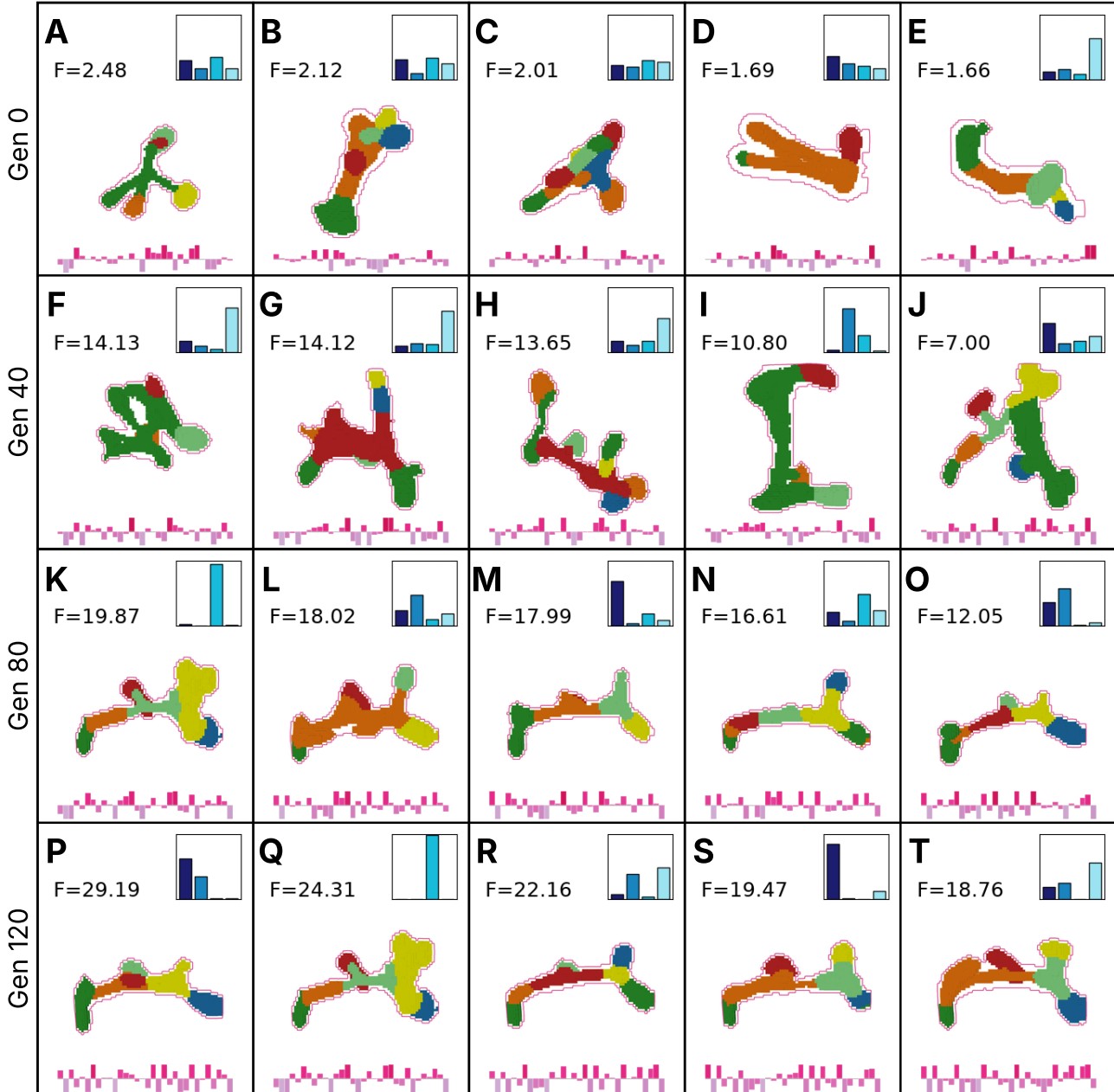

*Figure 3*. **Top five designs at different points in evolutionary time.** The best designs from a randomly initialized population (**A-E**), early (**F-J**) and late (**K-O**) in evolution, and from the final population (**P-T**) are shown for a representative Upright Locomotion trial. Each body plan contains an internal jointed skeleton (multicolored segments) surrounded by soft tissue (magenta line). In the top right hand corner of each design panel, an inset bar plot shows how weight was routed across the four experts (blue rectangles) to control the given body, with taller and shorter bars corresponding to more and less weight, respectively. Below each design, part of the genotype (every 16th element of the vector) is visualized by upward (dark pink) and downward (light pink) bars representing positive and negative values, respectively, and with bar length proportional to the absolute value of the component.

population about the path. Given that ECo-MoE does not appreciably alter the depth or breadth of design exploration, increases in evolvability appear to be due to better exploitation (more effective control) of the designs encountered during evolution. This may to some extent buffer deleterious mutations and thereby open otherwise intractable paths through design space.

### 3.2. Analysis and ablations.

Fig. 5 tracks expert usage within individual robots and across the population, over evolutionary time. As ECo-MoE dynamically adapts expert allocation, multiple experts continue to be used across the population, but different experts become increasingly specialized for specific structures.

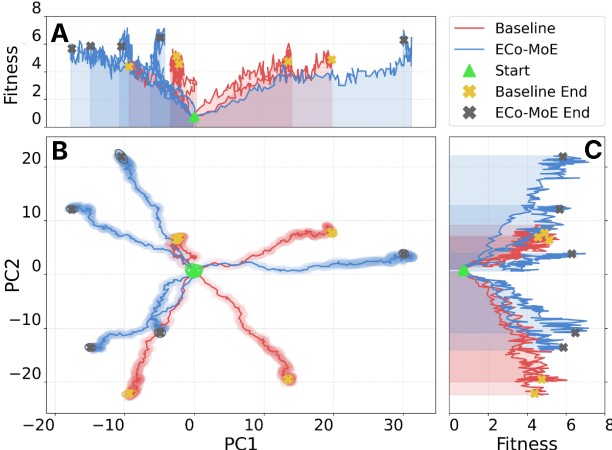

*Figure 4.* **Evolution in latent design space.** In order to visualize how evolution moved through the latent space, we performed PCA to obtain a 2D projection (**A-C**). The mean (lines in **B**) and standard deviation (shaded ellipses in **B**) of the evolving design populations are drawn along the first two principal components (PC1 and PC2). Five independent paired trials are shown for Upright Locomotion. In each trial, ECo-MoE (blue) and the baseline (red) begin from similar initial latent samples and population distributions (green triangles and ellipses) but terminate in distinct regions of the latent space (yellow and black Xs). These evolutionary paths are shown from three orthogonal views: fitness and PC1 (A), PC1 and PC2 (B), and fitness and PC2 (C). The same views of the other task environments can be found in Appx. Fig. 9.

Appx. Fig. 11 shows the results of an otherwise equivalent ablation study that removed routing diversity regularization. With regularization, expert usage remained distributed across multiple experts throughout evolution. Without regularization, routing quickly collapses to a single expert.

We also evaluated the sensitivity of ECo-MoE to the number of experts by running additional trials with two and eight experts (Fig. 5). To ensure a fair comparison with both the baseline, which can be viewed as a single expert, and the main ECo-MoE model with four experts, we adjusted the size of each expert so that all variants have a comparable overall parameter budget (see Table 2). ECo-MoE was found to be robust to the selected number of experts; all tested numbers of experts outperformed the non-modular (single expert) baseline (Appx. Fig. 10). With just two experts, both experts were used cooperatively across the population, but the routing patterns appeared to be less sensitive to morphological differences. With eight experts, only a subset of experts was consistently utilized, while the remain-

*Table 2.* Model size comparison.

| Experts | Policy size | Expert size | Critic size |
|---------|-------------|-------------|-------------|
| 1 | 2.563M | 2.563M | 2.038M |
| 2 | 2.337M | 1.168M | 2.038M |
| 4 | 2.388M | 0.597M | 2.038M |
| 8 | 2.439M | 0.304M | 2.038M |

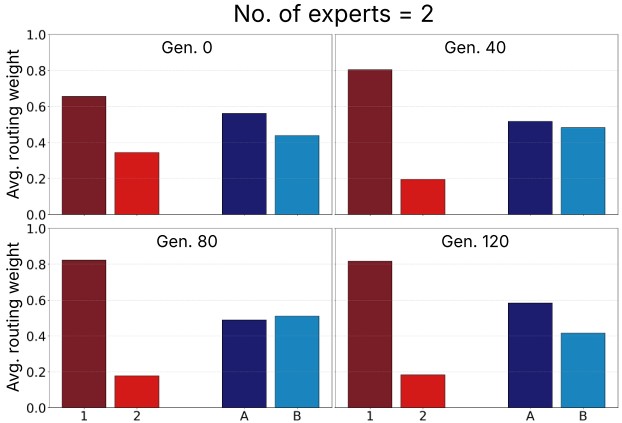

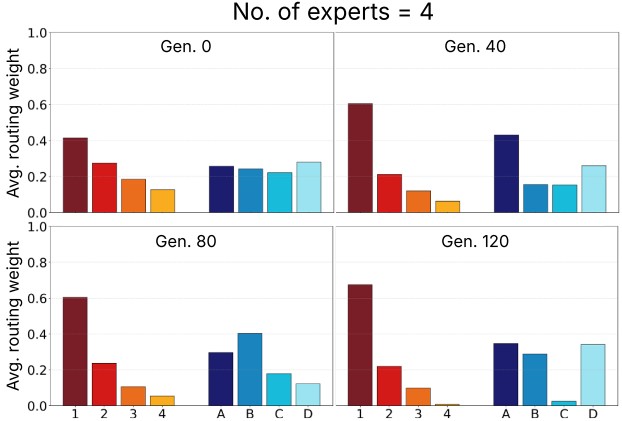

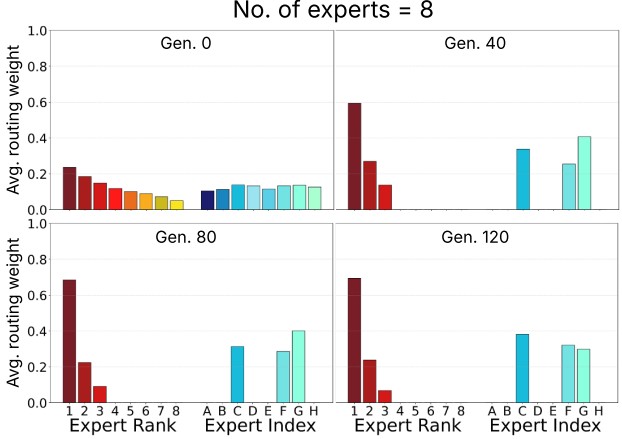

*Figure 5.* **Routing weight distribution vs. mixture size.** Before deciding to provide ECo-MoE a mixture comprising four experts in our experiments throughout this paper, we briefly explored different mixture sizes. Routing weight distribution is illustrated for a representative Upright Locomotion trial, at four points during evolution (gen 0, 40, 80 and 120), using two experts (top), four experts (middle), and eight experts (bottom). Reddish bars show the average routing weights for each design individually after sorting experts by weight rank in that generation. Bluish bars show the average routing weight assigned to each expert index across the entire population. We use A,B,...,H here to refer to the index, with A corresponding to $\pi_0$ and H corresponding to $\pi_7$.

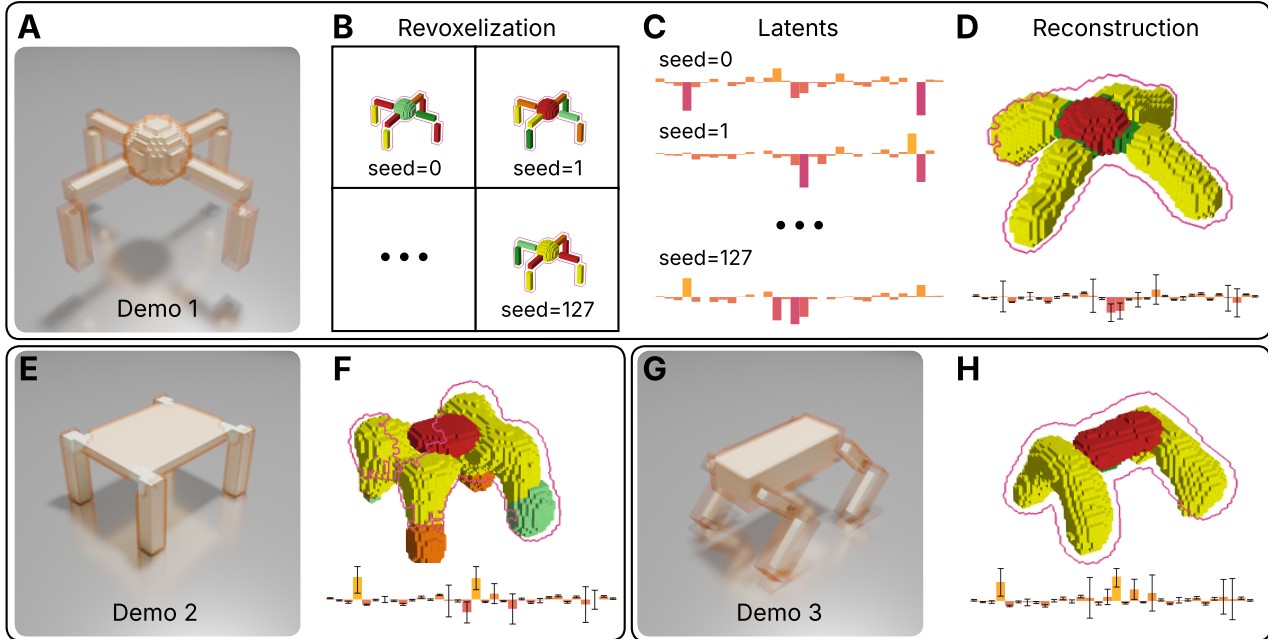

*Figure 6.* **Converting a predefined design into a latent prior.** The predesigned demo (**A**) was revoxelized to match the $64^3$ resolution of the encoder/decoder. This step was repeated 127 times to create 128 revoxelizations, each with a different random reindexing of the bones within its skeletal graph (**B**), which were encoded to generate a collection of 128 latent genotype vectors (**C**; for visualization, only 32 of the 512 latent dimensions are shown). The mean and standard deviation of this collection define the initial design distribution prior to evolution. Decoding the mean genotype at the center of the distribution reconstructs a morphology (**D**) that is structurally similar to the original. The same process was used to encode and decode other demos into genotypes and back into phenotypes (**E-H**). Although the decoded designs do not perfectly reconstruct the demos, they can nevertheless provide a useful prior for evolution.

ing experts received near-zero routing weight throughout evolution. Under the tested settings, four experts provided the best balance among task performance, specialization, and effective parameter usage.

### 3.3. Evo by demo.

Next, we study the potential of ECo-MoE to steer evolution under three settings. The first, which we refer to as "PretrainOnly", injects a pretrained expert controller (for a predesigned demo) into the mixture and keeps its parameters frozen throughout evolution. The second setting, which we call "PredesignOnly", initializes evolution with a prior distribution centered about the aggregated latent genotypes of a predesigned demo (Fig. 6). The third, "CoSteering" setting combines the settings of the first two, utilizing both a frozen pretrained expert and a predesigned initialization.

We considered three different predesigned demos: an 8-DoF radially symmetrical quadruped (Demo 1; Fig. 6A), a 4-DoF bilaterally symmetrical quadruped (Demo 2; Fig. 6E), and an 8-DoF bilaterally symmetrical quadruped (Demo 3; Fig. 6G). An expert policy was separately trained for each. To ensure policy expertise, a thorough hyperparameter sweep was conducted (see Appx. B for details).

Evo-by-demo with a predesigned initialization (CoSteer-

ing and PredesignOnly) achieved significantly higher fitness than the basic, randomly initialized ECo-MoE and its non-modular baseline (Figs. 8G, 12C and 13C). Providing a pretrained expert in addition to predesigned initialization (CoSteering) produced evolved morphologies with the strongest resemblance to demo, in terms of skeletal topology and bone geometries.

More concretely, CoSteering was the most effective at steering evolution toward designs with morphological metrics (defined below) close to those of the first demo (Fig. 6A), and it did so more reliably than PretrainOnly and PredesignOnly (Fig. 8E and F). The same seems to be true for the second demo (Fig. 6E) but not the third (Fig. 6G). See Appx. Figs. 12 and 13. For the third demo, PredesignOnly pulled evolution closest into morphological alignment with the demo (Appx. Fig. 13A and B).

**Morphological metrics.** To quantify the steerability of evolution under these different conditions we first identified the root segment. Let $\mathbf{x}_i$ denote the CoM (center of mass) of bone $i$ and let $m_i$ denote its mass. The root segment is the bone with minimal distance to the overall CoM of the whole body:

$$\arg\min_i \left\| \mathbf{x}_i - \frac{1}{\sum_i m_i} \sum_i m_i \mathbf{x}_i \right\|_2 \quad (10)$$

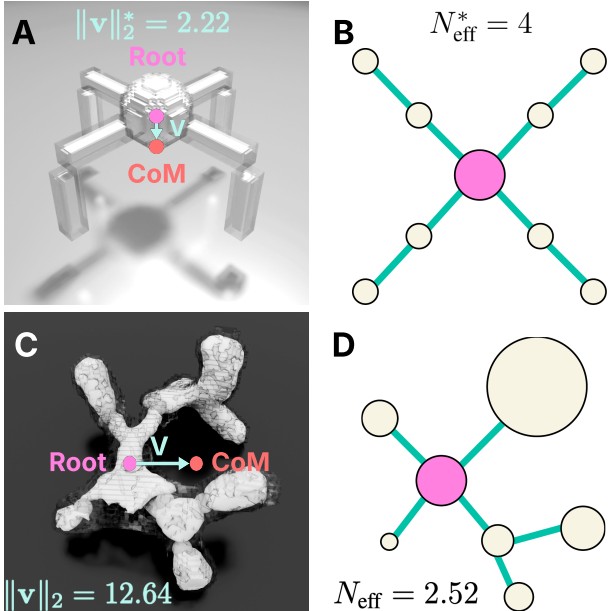

*Figure 7.* **Morphological metrics.** The skeletal root (i.e. the center of the spherical torso) of the symmetrical demo (**A**) stands slightly above its CoM, corresponding to a relatively small mass-bias-vector-magnitude of $\|\mathbf{v}\|_2^* = 2.22$. Its skeletal graph has four balanced branches (**B**), an effective-limb-count of $N_{\text{eff}}^* = 4$. The asymmetrical body of the evolved robot (in **C**), by contrast, has a much larger mass-bias of $\|\mathbf{v}\|_2 = 12.64$, and because of its uneven branching (**D**), which includes a dominant skeletal branch, its effective-limb-count is just $N_{\text{eff}} = 2.52$.

We then construct a rooted spanning tree utilizing the already filtered, cycle-free rigid graph topology and treat each neighbor of the root as a subtree branch. Letting $M_j$ denote the total mass of branch $j$, and normalizing branch mass by the total non-root mass $M_{\text{nr}} = \sum_j M_j$, we obtain the branch mass fraction $q_j$:

$$q_j = \frac{M_j}{M_{\text{nr}} + \varepsilon}, \qquad \sum_j q_j \approx 1, \qquad (11)$$

where $\varepsilon$ is a small constant for numerical stability. Finally, we define the *effective-limb-count* as

$$N_{\text{eff}} = \frac{1}{\sum_j q_j^2 + \varepsilon}, \qquad (12)$$

which is maximized when the mass is evenly distributed across many branches and approaches $1$ when a single branch dominates. Since all three of our demos have four limbs with identical shape and uniform mass, the reference value of the effective-limb-count is $N_{\text{eff}}^* = 4$.

We define a second metric, the *mass-bias-vector-magnitude*, which captures asymmetries in the robot's mass distribution relative to its root segment:

$$\mathbf{v} = \mathbf{x}_{\text{CoM}} - \mathbf{x}_{\text{root}} = \frac{1}{M} \sum_i m_i (\mathbf{x}_i - \mathbf{x}_{\text{root}}), \quad (13)$$

where $m_i$ is the mass of bone $i$, $\mathbf{x}_i$ is its CoM, $\mathbf{x}_{\text{root}}$ is the root position, and $M = \sum_i m_i$ is the total body mass. A Euclidean norm of this vector yields a scalar metric:

$$\|\mathbf{v}\|_2 = \sqrt{v_x^2 + v_y^2 + v_z^2}. \qquad (14)$$

According to these metrics (Fig. 7) robots evolved under ECo-MoE not only possess distinct genotypes compared to those evolved under the baseline (Fig. 4B), the expression of those genotypes produced more balanced anatomies with higher effective-limb-counts (Fig. 8E) and lower mass-bias-vector-magnitudes (Fig. 8F). This implies that ECo-MoE does not merely improve the control of designs during evolution, it results in the evolution of more controllable designs.

## 4. Discussion

Descent from a common ancestor has resulted in a shared toolkit for control of morphology and behavior across animals as diverse as insects, fish, elephants, and humans—in these and nearly every other animal, the same modular genes generate the same basic body parts (Pearson et al., 2005) and the same modular neural circuits produce the same basic movement patterns (Ijspeert, 2008), as just two examples. In this paper, we presented a model of evolution and learning in robots that provides a high level abstraction of these modular cross-embodiment control systems. More specifically, we jointly co-optimized a population of latent genotypes that determine morphology and a mixture of neural experts that drive behavior. The same modular genes and modular neural circuits were thus shared by all robots.

Our results suggest that this framework can increase the evolvability of robots (Fig. 2), particularly as the complexity of the behavior (Fig. 2B) or environment (Fig. 2C) increases. It also admits "evo by demo" (Fig. 8), a novel approach to steerable co-optimization of morphology and control. Following Li et al. (2025), our robots combine freeform skeletons and soft tissues in an open-ended design space. However, evolution was constrained to a relatively small subspace, namely the VAE latent manifold.

Although the VAE could partially reconstruct (more or less) the predesigned quadrupedal demos (Fig. 6), it failed to adequately reconstruct familiar bipedal and hexapodal forms (Appx. Fig. 14). This is likely due to how synthetic data (example body plans) were generated during VAE training. The VAE could be retrained using a more diverse dataset that includes bipedal and hexpodal examples. But any static compression of a solution space will struggle to reconstruct or generate certain out-of-distribution structures. Future work could address this problem by adapting the VAE and thereby expanding or reshaping the latent manifold incrementally over the course of evolution, e.g. by updating the encoder and decoder (i.e. the whole genome) or only the

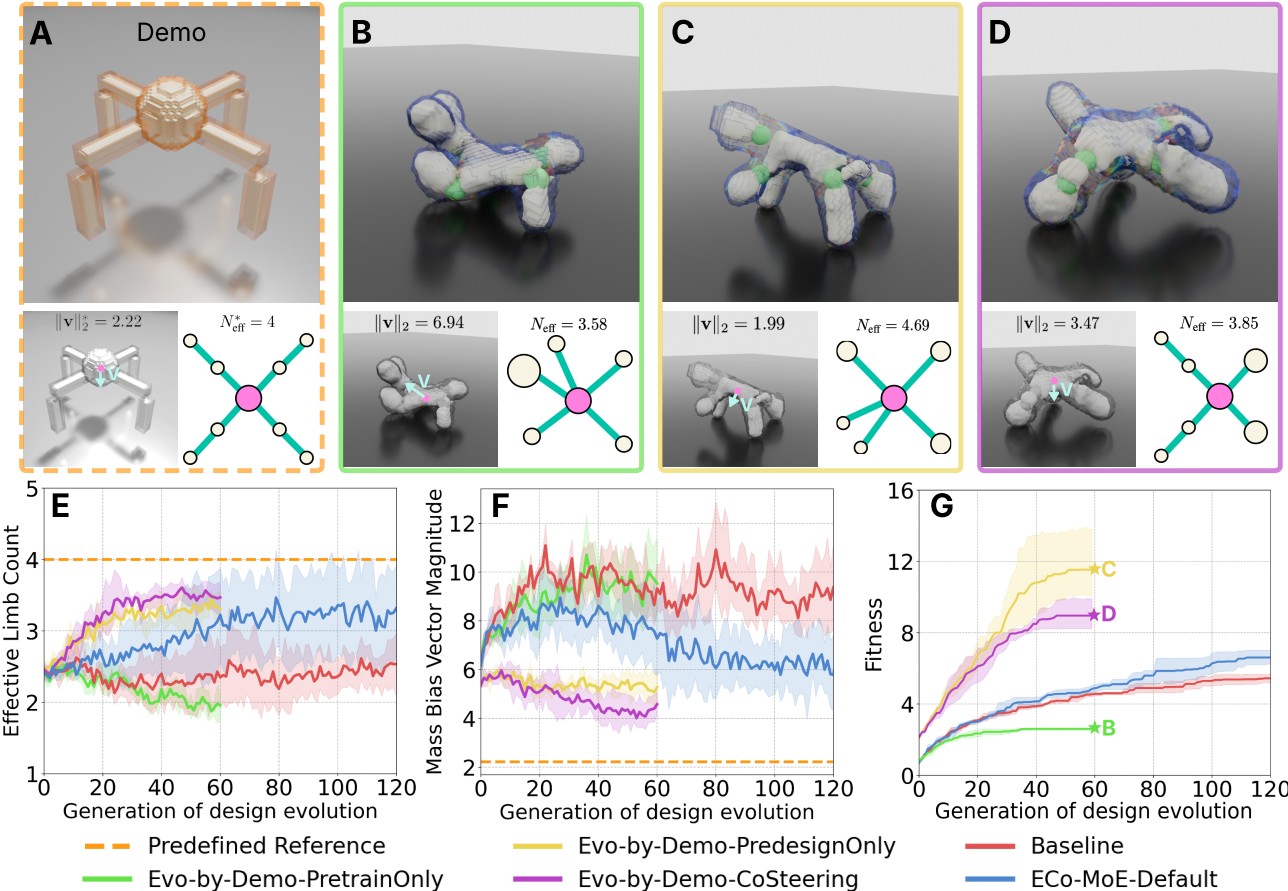

*Figure 8.* **Evo by demo.** A predesigned body plan (the demo; **A**) and its pretrained expert policy were used to initialize and guide evolution toward morphologies with similar phenotypic traits. A representative evolved morphology is shown for each of the three tested variants of evo-by-demo: without a predesigned latent initialization (**B**; PretrainOnly), without a frozen pretrained expert (**D**; PredesignOnly), and with both predesigning and pretraining (**D**; CoSteering). Five independent evolutionary trials of each variant were conducted for Upright Locomotion and compared against each other as well as the basic, randomly initialized ECo-MoE and its non-modular baseline. Two morphological metrics were used to track evolution relative to the demo: the effective-limb-count (**E**; defined in Sect. 3.3) and the mass-bias-vector-magnitude, which quantifies asymmetry in the robot's mass distribution relative to its root segment (**F**; see Sect. 3.3). Fitness was also tracked (**G**; averaged across the population as in Fig. 2). Under the tested conditions, CoSteering (purple lines in E-G) was the most effective way to guide evolution toward the demo's radially symmetrical quadrupedal form (orange dashed lines in E and F), but the highest fitness was achieved without pretraining (yellow line in G; PredesignOnly). Overall we see that a good demo, such as this one, can significantly increase the fitness of evolved designs, but only if initialized about the predesigned latent (G). Shaded regions indicate 95% bootstrapped confidence intervals of the mean. Results for other demos can be found in Appx. Figs. 12 and 13.

decoder (i.e. just the genotype-to-phenotype map), based on the most promising robots encountered during search.

Another assumption of the adopted VAE that caused unnecessary complexity here, specifically for evo-by-demo, was that the same phenotype could be mapped to many different genotypes. This was because the individual bones in the skeletal graph were assigned an arbitrary label (different colorings of the same body in Fig. 6B), and joints were then carved out between them. A simple fix would be to explicitly encode joints within the body, which would obviate the need for bone labels and greatly simplify predesign.

Finally, it is important to emphasize that the experiments in this paper were restricted to locomotion-based behaviors in simulated land-based environments. Though we demonstrated that ECo-MoE can enable cross-embodiment control for locomotion on complex terrain (Potholes), scaling to more diverse and more challenging environments, and doing so in physical machines (Guo et al., 2026), remains an important next step. Swimming, object manipulation, and other behaviors that are well suited for soft endoskeletal forms would be natural extensions.

An exciting area of future work would be to evolve robots that exhibit multiple heterogeneous behaviors. Perhaps this could be accomplished through hierarchical routing, where lower-level experts capture embodiment-dependent motor primitives and higher-level experts specialize in task objectives and sequencing.

## Impact Statement

This paper models the evolution of complex embodied agents in a simulated terrestrial environment. We hope this unique abstraction will eventually prove useful for thinking about and explaining how organisms evolve and learn in nature. This work could also have important practical implications for robotics (Lipson & Pollack, 2000; Matthews et al., 2023; Yu et al., 2026) as well as artificial life (Kriegman et al., 2020; 2021a), the benefits and dangers of which are well explored in the literature and popular culture. The robots in this paper are only just beginning to be realized (Guo et al., 2026), but they have the potential to be more beneficial and less dangerous than current rigid bodied forms because they are surrounded by soft tissues and evolved to be controllable.

## Acknowledgments

This research was supported by NSF awards 2331581 and 2440412, and TWCF award 20650.

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

## A. Task Setup

Table 3 below provides the hyperparameters we used for reinforcement learning and evolutionary strategies. The pretrained VAE checkpoint, as well as the compiler and validity checks used to decode latent vectors into simulatable morphologies, were adopted from Li et al. (2025). However, we found duplicating multiple "clones" per individual in the population to be unnecessary under the tested conditions. Without clones, nearly the same fitness was achieved using half the compute. We also observed that reducing the number of policy training epochs, and instead evaluating more designs, increased the fitness of the final population, even for the baseline setup. Finally, increasing the PPO training batch size while reducing the number of training steps per epoch led to more stable fitness improvements over evolutionary iterations (generations).

*Table 3.* Hyperparameters for learning and evolution.

| Parameter | Value |
|---|---|
| *Reinforcement Learning* | |
| Actor learning rate | $6 \times 10^{-5}$ |
| Critic learning rate | $6 \times 10^{-5}$ |
| Discount ($\gamma$) | 0.9 |
| Entropy weight | 0.01 |
| GAE | Not used |
| Learning epochs | 5 |
| PPO train batch size | 128 |
| PPO train steps per epoch | 40 |
| *Evolution* | |
| Population size | 64 |
| Cloned instances of each design | 1 (i.e. no clones) |
| Top designs preserved from previous gen | 4 |

## B. Pretraining Experts

For training the set of experts tailored to predesigned demos, we followed the standard quadruped gait-training practice of shaping rewards with a small number of interpretable terms such as movement, posture, height, and action smoothness.

**Expert objective.** For each demo, we trained the expert policy to maximize a weighted sum of reward components,

$$r_t = \mathbf{w}^\top \mathbf{c}_t = w_{\text{move}} c_t^{\text{move}} + w_{\text{stand}} c_t^{\text{stand}} + w_{\text{height}} c_t^{\text{height}} + w_{\text{act}} c_t^{\text{act}}, \tag{15}$$

where $\mathbf{c}_t = [c_t^{\text{move}}, c_t^{\text{stand}}, c_t^{\text{height}}, c_t^{\text{act}}]$ collects per-step components and $\mathbf{w}$ denotes their weights. Here, we reuse the symbols $\mathbf{c}$ and $\mathbf{w}$ locally to refer to the reward components and their corresponding weights, respectively.

**Reward components.** Let $\mathbf{x}_t$ denote the center-of-mass (CoM) position of the torso at step $t$, $\hat{\mathbf{d}}$ the commanded movement direction (set to $+X$ by default, and body plan is rotated to face $+X$ according to its nominal movement direction), $\Delta t$ the control timestep, $\mathbf{u}_t^{\text{body}}$ the body "up" direction (e.g. torso frame $+Z$), $\mathbf{u}^{\text{world}}$ the world up direction, $h_t$ a representative body height (we use the nominal torso CoM height), and $a_t$ the action vector. The reward components were defined as:

$$c_t^{\text{move}} = \frac{(\mathbf{x}_{t+1} - \mathbf{x}_t)^\top \hat{\mathbf{d}}}{\Delta t}, \tag{16}$$

$$c_t^{\text{stand}} = \mathbb{I}\left[\langle \mathbf{u}_t^{\text{body}}, \mathbf{u}^{\text{world}} \rangle \geq \eta_{\cos}\right], \tag{17}$$

$$c_t^{\text{height}} = \mathbb{I}[h_t \geq h_{\min}], \tag{18}$$

$$c_t^{\text{act}} = \begin{cases} 0, & t = 0, \\ -\|a_t - a_{t-1}\|_2^2, & t > 0. \end{cases} \tag{19}$$

Here $\eta_{\cos}$ is a cosine threshold controlling how upright the torso must remain, and $h_{\min}$ is a minimum-height threshold to prevent crouching or collapse. We set $h_{\min}$ to 0.15m by default, roughly 0.75 of the starting $Z$ height of the torso CoM of

all three predesigned demos. We found this compact set of terms sufficient to produce robust forward locomotion while discouraging excessive tilting and uneven movement patterns.

**Reward-weight sweep.** Because the appropriate balance between posture/height regularization and agility depends on the morphology, we tuned $\mathbf{w}$ via a small grid search (Table 4). Concretely, we fixed $w_{\text{move}} = 1.0$ and swept the remaining weights $w_{\text{stand}}, w_{\text{height}}, w_{\text{act}}$. For each setting, we trained the expert with the same PPO implementation as in the main experiments, and selected the final configuration by the best validation performance (average episodic return and qualitative stability, i.e. sustained forward motion without frequent falls). We found adding explicit uprightness and height constraints to be particularly important for the chosen demos. A small action-smoothness penalty was added to improve gait consistency.

*Table 4.* Pretrained expert reward components and swept weight ranges for the predesigned demos.

| Component | Definition | Weight |
|---|---|---|
| Movement | $c_t^{\text{move}} = ((\mathbf{p}_{t+1} - \mathbf{p}_t)^\top \hat{\mathbf{d}})/\Delta t$ | $w_{\text{move}} = 1.0$ (fixed) |
| Uprightness | $c_t^{\text{stand}} = \mathbb{I}[\langle \mathbf{u}_t^{\text{body}}, \mathbf{u}^{\text{world}} \rangle \geq \eta_{\cos}]$ | $\{0.005, 0.01, 0.02\}$ |
| Height | $c_t^{\text{height}} = \mathbb{I}[h_t \geq h_{\min}]$ | $\{0.005, 0.01, 0.02\}$ |
| Action smoothness | $c_t^{\text{act}} = -\|a_t - a_{t-1}\|_2^2$ | $\{10^{-4}, 5 \times 10^{-4}\}$ |

**PPO hyperparameter sweep.** In addition to reward-weight tuning (Table 4), we performed a lightweight grid search over PPO optimization hyperparameters to obtain stable expert training across predesigned demos (Table 5). Concretely, we swept actor/critic learning rates and the GAE parameter $\lambda$, while keeping the discount factor fixed at 0.99 to encourage temporally consistent behavior over an episode. We set the entropy regularization weight to $-0.01$ (i.e. an explicit entropy bonus) to promote exploration and discovery of well coordinated, posture-stable gaits. We selected the final settings by validation performance and training stability (i.e. consistent return improvement without collapse across seeds) over the same PPO training batch size, steps per epoch, and total epochs as the base tasks.

*Table 5.* PPO hyperparameter sweep for expert pretraining.

| Parameter | Values |
|---|---|
| Actor learning rate ($\alpha_{\text{actor}}$) | $\{10^{-3}, 3 \times 10^{-4}, 10^{-4}\}$ |
| Critic learning rate ($\alpha_{\text{critic}}$) | $\{10^{-3}, 3 \times 10^{-4}, 10^{-4}\}$ |
| GAE ($\lambda$) | $\{0.8, 0.9, 0.95, \text{Not used}\}$ |
| Discount ($\gamma$) | 0.99 (fixed) |
| Entropy weight ($\beta_{\text{ent}}$) | $-0.01$ (fixed) |

We report in Table 6 below the best hyperparameters we found for the predesigned demos.

*Table 6.* Selected hyperparameters for all predesigned demos.

| Swept parameter | Best value |
|---|---|
| Actor learning rate ($\alpha_{\text{actor}}$) | $6 \times 10^{-5}$ |
| Critic learning rate ($\alpha_{\text{critic}}$) | $6 \times 10^{-5}$ |
| GAE ($\lambda$) | 0.95 |
| Entropy weight ($\beta_{\text{ent}}$) | $-0.01$ |
| Discount ($\gamma$) | 0.99 |
| Reward standing weight ($w_{\text{stand}}$) | 0.005 |
| Reward height weight ($w_{\text{height}}$) | 0.005 |
| Reward action-smoothness weight ($w_{\text{act}}$) | $10^{-4}$ |

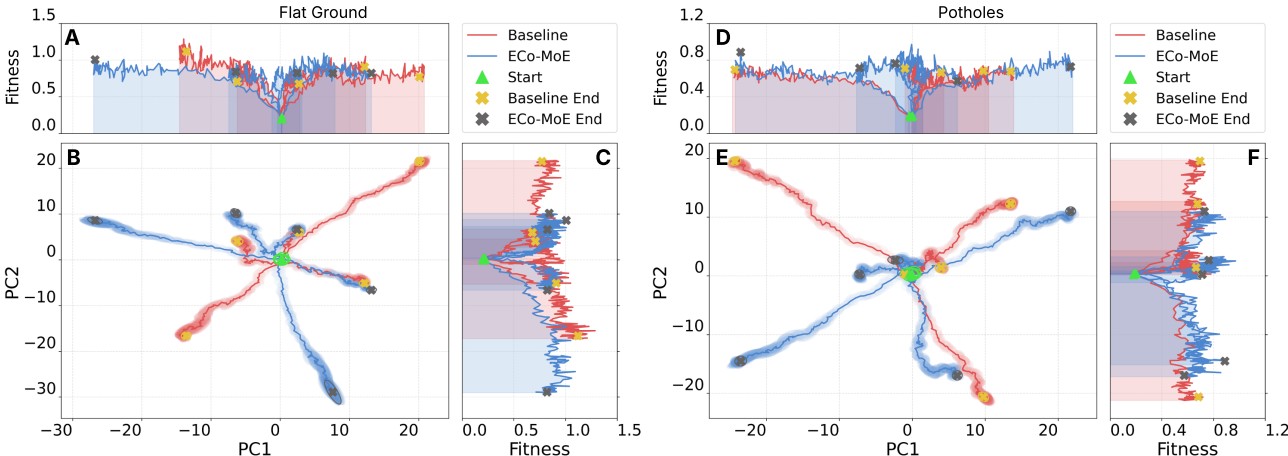

*Figure 9.* **Evolutionary dynamics on Flat Ground and Potholes.** This is a recreation of Fig. 4 for the Flat Ground (**A-C**) and Potholes (**D-F**) tasks. See Fig. 4 in the main text for details. In brief, evolutionary paths are shown across a 2D projection of latent space (B; E) and along the resulting fitness landscape (A and C; D and F).

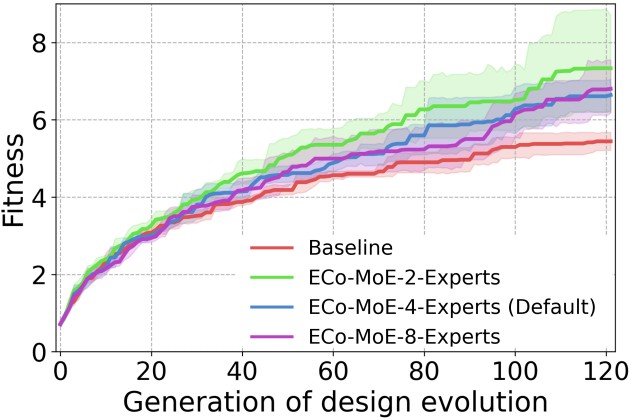

*Figure 10.* **Does the number of experts matter?** In terms of beating the nonmodular (single expert) baseline, no. More specifically, we compared mixtures comprising two, four, and eight experts across five independent evolutionary trials for Upright Locomotion. In each case, expert size was adjusted so that total model size remained comparable (Table 2). Shaded regions indicate 95% CIs. See Fig. 5 for expert usage over evolutionary time.

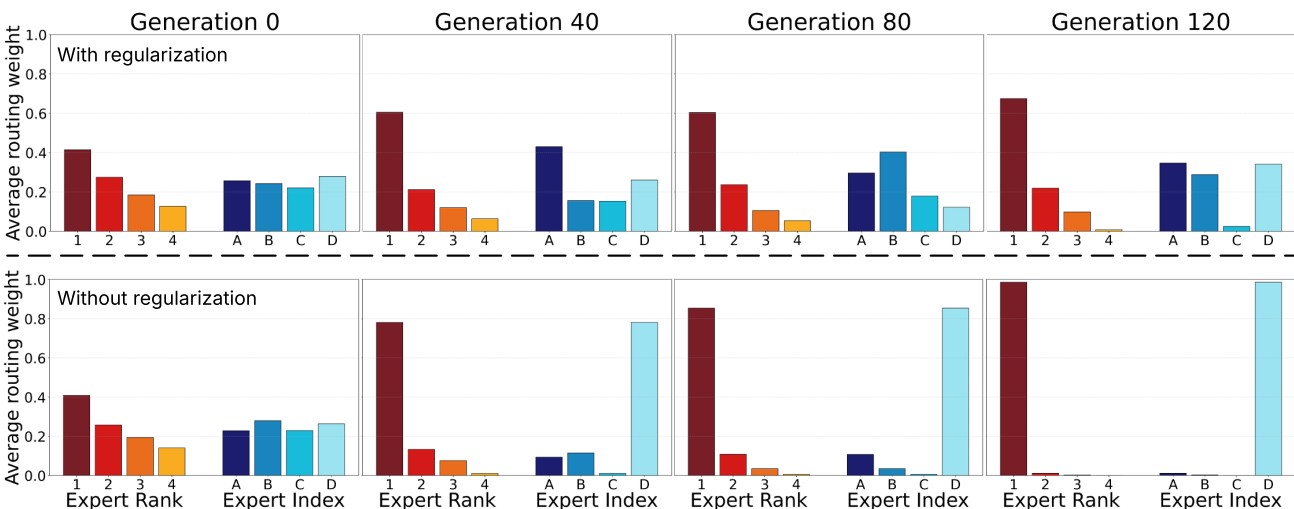

*Figure 11.* **Does routing diversity regularization matter?** Yes. Without it, routing weight becomes increasing concentrated on a single expert, resulting in expert collapse. The top panels show how routing weight distributions change over evolutionary time with routing diversity regularization. The bottom panels show the same, but without routing diversity regularization. Reddish bars show the average routing weights for each design individually after sorting experts by weight rank in that generation. Bluish bars show the average routing weight assigned to each expert index across the entire population. We use A,B,C,D here (as in Fig. 5) to refer to the index, with A corresponding to $\pi_0$ and D corresponding to $\pi_3$.

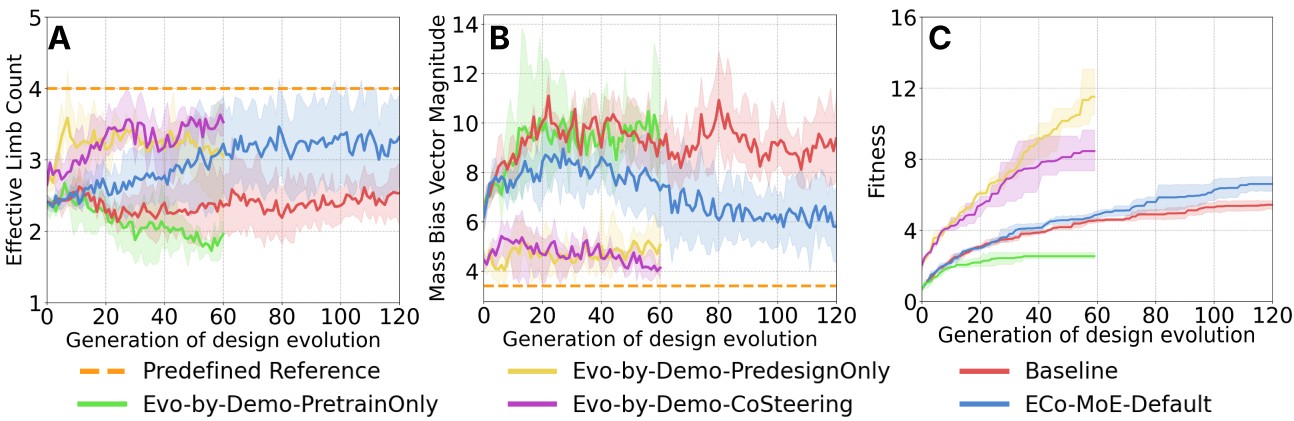

*Figure 12.* **Evo by Demo on Demo 2.** This is a recreation of Fig. 8E-G for the second demo we tested, which is shown in Fig 6E.

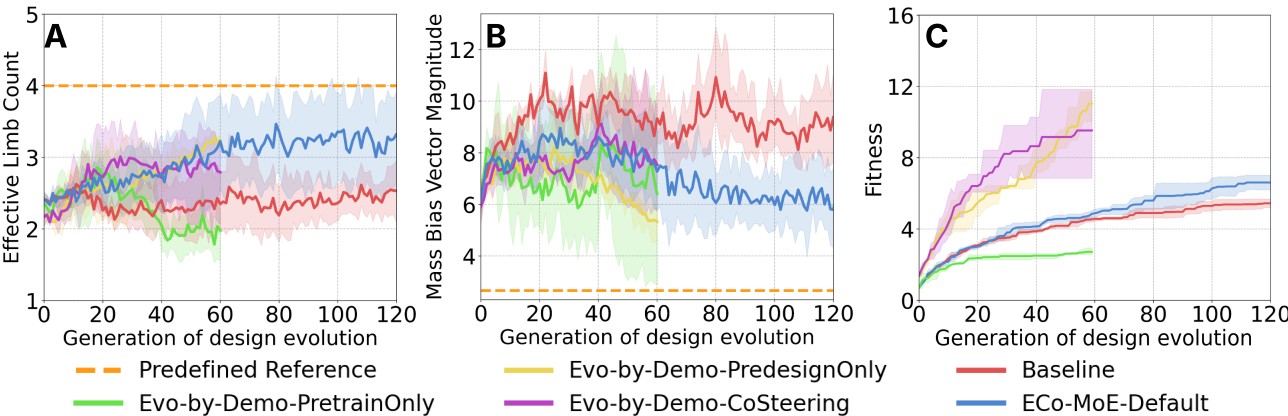

*Figure 13.* **Evo by Demo on Demo 3.** This is a recreation of Fig. 8E-G for the third demo, which is shown in Fig 6H.

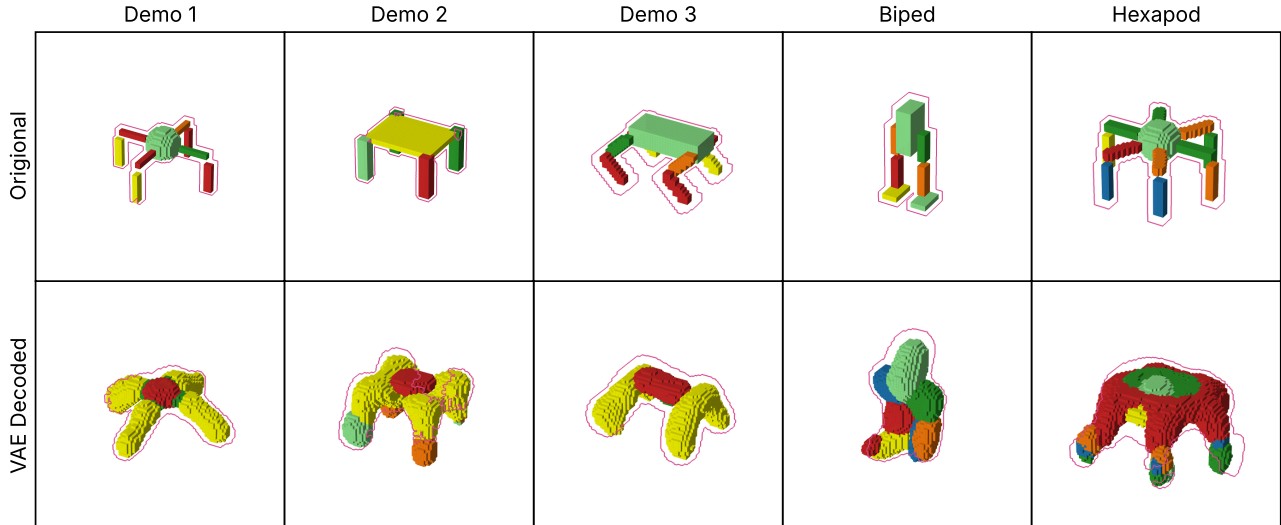

*Figure 14.* **Visualizing reconstruction loss of the adopted VAE.** Top row shows the original reference morphologies: the three quadruped-style demo demos used in our main experiments, alongside a biped and a hexapod. Bottom row shows their reconstructions after VAE encoding and decoding.

