# OpenReview forum: "ECo-MoE: Embodiment-Conditioned Mixture of Experts Increases the Evolvability of Robots"
_ICML.cc/2026/Conference — ICML 2026 regular_

### Official Review · Reviewer_Q1AV · 2026-02-19

**Soundness:** 3
**Presentation:** 3
**Significance:** 2
**Originality:** 3
**Overall Recommendation:** 4
**Confidence:** 3

**Summary:**

To my understanding, the contribution of this paper is the novel integration of a mixture of experts controller which is co-optimised alongside an evolving robot morphology. Prior work considered either one controller per population member, a single controller for the whole population, or a single controller for the whole population conditioned on information about the morphology. This work adds an alternative method for the last case, a mixture of experts. They also show how their approach enables a novel paradigm of "evo by demo" where one of the mixture of experts is a pretrained controller, or the initial latent distribution can be shaped by the pretrained morphologies.

**Compliance With Llm Reviewing Policy:**

Affirmed.

**Final Justification:**

The authors presented new evidence during rebuttal which supported their contribution.

**Key Questions For Authors:**

1. How does exploration change with the number of experts, K?
2. What limb count / performance does the 'unguided' evolution reach?
3. Are there any other metrics (or additional statistical analysis) which could make the improvement of 'co-steering' more clear?

**Limitations:**

Yes

**Strengths And Weaknesses:**

The strengths of this paper include the clear presentation, high quality figures, interesting and well justified method, strong literature review, and good evaluation. I liked the motivation behind the mixture of experts, as well as the idea of steering evolution using an expert controller. The PCA analysis of exploration was very interesting. and the improvements to fitness were impressive. I thank the authors for their submission, which I enjoyed reading. I hope that my comments are helpful for improving the paper.

The main weakness of this paper is some lack of clarity around the contribution, and how the evaluation is demonstrating it. Here are some more detailed comments:
* There is no statement of contribution, I would greatly recommend adding one, as it is crucial for readers to clearly understand what the contribution is. My main comments refer to my interpretation of your contribution, so clarity here would improve the paper.
* The first part of the contribution is clear - the mixture of experts controller is added to improve the evolvability. This is then supported by the results in Figure 2 and Figure 4. It is clearly discussed in the literature review and contrasted with prior work. I would just recommend that the evaluation of this contribution be improved slightly:
	* The evaluation and choice of baseline is good. However, to make the evaluation more convincing, it would be great to see the PCA analysis on more environments. The plots of improvement and displacement could be across multiple environments. Then, some statistical analysis could be included to determine if the change is statistically significant.
	* Since the method of Li et al. 2025 already has a mechanism for conditioning the controller on the geometry of the robot, it is not completely clear how the mixture of experts differs, beyond an architectural change. I think it would improve the paper to emphasise both in the text and the evaluation what differences your method has introduced. To support that, I think an ablation over the number of experts would be important (for example K=1 (Li et al), K=2, K=4, K=8). This would clearly show the reader how the mixture of experts mechanism was changing the exploration.
* The second novel contribution, directed evolution, is less clear in the text and in the evaluation. I recommend further discussion of this in the introduction and related work, as well as some expansions to the evaluation:
	* I would strongly recommend mentioning this more in the introduction and literature review to highlight the critical gap, because this is an important aspect of your contribution. It is important for the reader to understand what is novel about this, why it couldn't be done before, and why it is important.
	* The evaluation of evo by demo was good, but I had some questions as a reader. I would suggest to more clearly structure the evaluation to make it easier to follow. In general, choose the metrics which demonstrate clearly how evo by demo is working, and add comparison for the case where there is no demo - what does evo by demo enable which was not possible before. What limb count does the 'undirected' evolution achieve? Are there additional metrics which can support how evo by demo is guiding evoluation? Are there any relevant baselines? These points would help the reader clearly see how the contribution is supported by the evaluation. Partly, additional discussion of the critical gap can also answer these questions.
	* Paragraph at line 397 column 1 starting: "The quantitative trends...". This paragraph raises several good points, 1) co-steering drives the effective limb count towards the reference, and 2) co-steering was more stable. However, as a reader, I was not sure how these statements were supported by the data. I would highly recommend inserting numerical analysis that supports these claims and makes them more quantitative. Looking only at Figure 5, I was not sure whether these claims were supported. A mathematical backing would help, for example, 'across the three designs, the effective limb count for co-steering increased by 20% by generation 60, compared to 10% with design only and -5% with pretrain only. Regarding stability, co-steering averaged a limb count increase of $0.6 \pm 1.5$ per step, a 15% lower standard deviation than next most stable.' Some quantitative analysis like this would really help the reader to understand the core message.
	* Related very strongly to the previous point, the reader wants to understand whether the improvements are significant or not. The evaluation of directed evolution would be improved by explaining how many seeds were run, it appears from the current text to be only one. If it is only one, then it isn't clear whether these are statistically significant results, given the high variability. It would improve and strengthen this evaluation to include more seeds, and add statistical analysis.
	* A small question, does the effective limb count = 4 for the references come from using equation (12)? Since their masses are distributed differently I wondered what the limb count from equation (12) would come to. If the value $N_{eff}=4$ does come from equation (12), I suggest adding that small detail to the text. If not, does it represent a fair comparison? In that case, I would also suggest justifying why the comparison is/isn't fair in the text.
* A minor point, at line 204 column 1 (start of results), question 3: "Which design choices impact the performance of ECoMoE?". I was not sure where this question was answered, or whether it corresponded to an evaluation experiment. As I mentioned previously, it would be great to have an ablation over the number of experts, as this is a key component of your method.

---

> ### Author Rebuttal · Authors · 2026-03-31
>
> We appreciate the reviewer’s thoughtful feedback on both the paper’s analysis and organization. In response, we have added new experiments and analyses to the paper, and clarified the contribution. Please see Figs. S1-S9 in our anonymous supplementary materials, here: https://inspiring-cendol-ab20d0.netlify.app.
> ## PCA analysis on more environments.
> To better understand the behavior of ECo-MoE, we tracked the movement of the evolved population through the PCA-projected latent space in Figs. S2-S4 for all tasks as requested, including per-trial PC displacement and traversed area bar plots across five paired independent trials per task. In response to Reviewer t7XA, who identified an issue with our initial PCA projection (prefiltering latent dimensions by their correlation with fitness injects bias that distorts the visualization), we reperformed this analysis across all tasks using an unbiased PCA over all 512 dimensions of latent space. For comparison, in case the reviewer is curious, we provide the original, biased analysis using the top 100 fitness-correlated dimensions; however we plan to exclude the biased projections from our revised manuscript as they are somewhat misleading. In light of this new analysis, the advantage of ECo-MoE does not appear to result from exploring farther into latent space, but rather from optimizing more relevant dimensions.
> ## Are there any additional metrics for interpreting the evolution process?
> Characterizing how freeform volumetric designs change over time is difficult, but we find that defining shape metrics that explain how fitness is optimized is useful. Across all evo-by-demo experiments, whether augmented or not, we use the same base fitness function, which rewards morphologies that move as far as possible while maintaining as little contact surface as possible. We observe two solution patterns that may emerge in each trial: 1) standing higher with fewer legs to reduce contact while maintaining stability, and 2) standing lower with more legs for faster locomotion. We define a simple metric, called the mass bias vector magnitude, with the definition provided in the section 3.1 of the supplementary material. Intuitively, this metric captures how far the body weight is offset from the body core, and should have a negative correspondence with the effective limb count metric.
> ## How does expert number affect the ECo-MoE exploration?
> To measure how the number of experts affects exploration, we used the metric described above together with the previously defined limb count metric to capture the dynamics of evolution. We report the results in Fig. S6(A-C). This comparison study spans K=2, K=4, and K=8 (with the Baseline corresponding to K=1, i.e., Li et al.), directly illustrating how the mixture of experts mechanism changes exploration behavior beyond a purely architectural difference, and serves as our evaluation for Q3 ("Which design choices impact the performance of ECo-MoE?"). In general, while the ECo-MoE setup alone and the variation in expert number do not deterministically drive exploration toward either of the two solution patterns mentioned above, we observe that they do cause the results to converge more quickly toward one side. In contrast, the baseline model tends to straddle both ends and exhibits a more chaotic exploration process.
> ## Extended evaluation of Evo by Demo.
> In Fig. S7, we report both the effective limb count and the mass bias vector magnitude for the final evolved populations at generation 60, including the baseline model and default ECo-MoE as unguided references. Unguided evolution yields average effective limb counts of 1.843 (Baseline) and 2.389 (ECo-MoE-Default), far below the predefined reference of 4. Co-steering and design-only reduce the average error to this reference by 43.8% and 48.0%, respectively, while pretrain-only increases it by 20.8%. To establish statistical stability, we conducted repeated trials on predefined Design 1 in Fig. S8: three independent runs each for the Evo by Demo variants and five for the Baseline and default ECo-MoE. The morphology steering trends from Fig. 5 in our original submission are confirmed to be robust across seeds in Fig. S8(A, B). Importantly, Fig. S8(C) shows that co-steering and design-only both substantially outperform the Baseline and default ECo-MoE in raw task fitness with 95% bootstrapped confidence intervals, demonstrating that Evo by Demo does not recover predefined traits at the expense of locomotion capability.
> ## “A small question, does the effective limb count = 4 for the references come from using equation…”:
> Yes, the reference value of 4 is computed by applying equation (12) to the predefined morphologies, which have four limbs of identical shape and uniform mass. We will add this detail to the text.
> ## References
> Li, M., Kong, L., and Kriegman, S. Generating freeform endoskeletal robots. In Proceedings of the International Conference on Learning Representations (ICLR), 2025.

---

> > ### Author Rebuttal · Reviewer_Q1AV · 2026-04-03
> >
> > I thank the authors for their very detailed responses to my concerns. The new analysis and addition of seeds/variance to the plots strengthens the contibution. I will increase my score.

---

### Official Review · Reviewer_ozBC · 2026-03-03

**Soundness:** 2
**Presentation:** 3
**Significance:** 3
**Originality:** 3
**Overall Recommendation:** 4
**Confidence:** 4

**Summary:**

This paper proposes ECo-MoE (Embodiment-Conditioned Mixture of Experts), a modular control architecture for the joint co-optimization of robot morphology and control within an evolutionary framework. Built on the latent design genome from Li et al. (2025), which encodes high-resolution freeform robots into a continuous latent space, ECo-MoE replaces the monolithic universal controller with expert subpolicies whose outputs are weighted by a linear gate conditioned on each robot's latent genotype.

The paper contributes in two aspects. First, it shows ECo-MoE improves evolvability over the single-policy baseline, particularly when the task demands morphological diversity (Upright Locomotion and Potholes environments), while using comparable or fewer parameters. Second, it introduces "evo by demo", which injects a frozen pretrained expert and/or a latent prior from a predefined design to steer evolution toward target morphological traits. Experiments on three quadruped-like reference designs show that combining both signals ("co-steering") most consistently guides evolved morphologies toward the desired structures.

**Compliance With Llm Reviewing Policy:**

Affirmed.

**Final Justification:**

The rebuttal has addressed my main concerns. I would stick to my original assessment (weak accept).

**Key Questions For Authors:**

1. In Eq. 8, fitness is scaled by the pretrained expert's gate weight, so evolution effectively balances task performance against overall compatibility with the predefined design. This seems unable to identify which specific substructures of the predefined morphology are actually useful — and indeed, Figure 5 shows that even co-steering produces only rough resemblance to the references. Have the authors considered more fine-grained steering mechanisms that could selectively preserve beneficial structural primitives (e.g., limb topology) rather than relying on a scalar compatibility score?

2. All three predefined morphologies in "evo by demo" are quadruped variants with similar four-limbed structure. Is this constrained by the VAE's limited latent coverage? Have the authors attempted steering toward fundamentally different body plans (bipeds, hexapods, snake-like forms)?

3. Figure 3 shows that a single predefined design is revoxelized into 128 variants using "randomly indexed skeletal graphs." Could the authors elaborate on this process? Specifically, how does one predefined morphology give rise to 128 distinct voxelized representations?

**Limitations:**

yes

**Strengths And Weaknesses:**

**Strengths:**
1. **Clear and well-motivated problem formulation**.
The paper clearly identifies two extremes in existing morphology-control co-optimization approaches: (1) training an individual policy per robot, which is computationally prohibitive, and (2) training a single monolithic universal controller for all robots, which can be overly conservative and lead to premature convergence that limits morphological diversity. ECo-MoE occupies a principled middle ground between these two extremes, and this motivation is both intuitive and well-grounded. The related work section provides a thorough survey situating these trade-offs within the broader literature.
2. **Simple and elegant method design.** The core design of ECo-MoE is commendably simple: a linear gating layer (Eq. 3–4) combines multiple expert outputs based on the robot's latent genotype z, and a routing diversity regularization loss (Eq. 5) encourages distinct routing patterns for morphologically distant designs, mitigating expert collapse. The framework is conceptually clear, easy to implement, and extends naturally to the "evo by demo" setting where a frozen pretrained expert is injected into the mixture.
3. **Insightful analysis and fair experimental comparison.** The latent-space trajectory analysis in Figure 2 (net displacement and traversed area) provides useful intuition for how ECo-MoE enhances evolvability by exploring a broader region of the fitness-correlated latent subspace. The comparison is also carefully controlled: Table 1 shows that ECo-MoE uses comparable or even fewer total parameters (2.388M vs. 2.563M) than the baseline, with an identical critic architecture, ruling out the possibility that performance gains simply stem from increased model capacity.

**Weaknesses:**
1. **Insufficient analysis of key design choices.**
The paper fixes K=4 experts and α=2 across all experiments without providing any sensitivity analysis or ablation studies. Different numbers of experts could significantly affect performance, expert utilization, and the susceptibility to expert collapse. Similarly, the compatibility weight α in Eq. 8 directly controls how strongly the pretrained expert steers evolution, yet its impact is not explored. More importantly, Section 3 explicitly poses the question "Which design choices impact the performance of ECo-MoE?" as one of three guiding research questions, but this question is never systematically addressed in the subsequent experiments.
2. **Limited baselines.**
The paper only compares against the monolithic universal controller from Li et al. (2025). However, other methods that also explicitly condition the policy on morphology could similarly alleviate the conservatism of a single shared controller. For instance, the hypernetwork-based approach of Xiong et al. (2023) generates morphology-specific controller weights and could, in principle, achieve a similar effect of specialization. The paper should at minimum provide a more thorough conceptual discussion of why MoE-based modularization is preferable to such alternatives, rather than leaving the comparison to a brief mention in the related work.
3. **Lack of in-depth analysis of learned gating behavior.**
The paper emphasizes that the modularity of MoE allows different experts to be independently updated and specialized for different body plans, yet provides no empirical evidence that this actually occurs. Key questions remain unanswered: What is the actual utilization rate and degree of specialization of each expert? Are morphologically distinct regions of the latent space indeed routed to different experts? How effective is the routing diversity loss (Eq. 5) in practice — does it truly prevent expert collapse? Adding further analysis (e.g., expert weight distributions across the latent space) would substantially strengthen the paper's core motivation.
4. **Organizational suggestions.** The current introduction interleaves the core motivation (the spectrum from individual policies to monolithic universal controllers, and the proposed middle ground) with extended background on evolutionary robotics and related methods. This makes the main narrative harder to follow. I would suggest separating the less directly related background into a dedicated Related Work section.

---

> ### Author Rebuttal · Authors · 2026-03-31
>
> We appreciate the reviewer’s constructive feedback and their suggestions regarding additional analysis, baselines, and clarification. Below we address the reviewer’s concerns and describe the additional experiments and analysis we conducted to answer the reviewer’s questions. Please see Figs. S1-S9 in our anonymous supplementary materials, here: https://inspiring-cendol-ab20d0.netlify.app.
> ## How do the number of experts, routing diversity regularization, and α affect ECo-MoE’s performance and gating behavior?
> We conducted several new experiments to answer the reviewer’s questions about expert utilization, specialization, and collapse prevention. To better understand the sensitivity of our results to the number of experts, we conducted new experiments with half as many (two) and twice as many (eight) experts, and tracked how expert usage and routing changed over evolution (Fig. S6). To better understand the effect of routing diversity regularization, we removed it in an ablation study (Fig. S5). Please also see our response to Reviewer Aprb for a summary of our findings. For the selection of α value, we chose 2 because when α=0, it is equivalent to the base unaugmented fitness; when α=1, we did not observe a strong steering effect that diverges morphology metrics from the baseline; and when  α=3, this caused the evolution of large, immobile morphologies with low fitness. We will add this justification to our revised manuscript.
> ## Is Evo by Demo constrained by the VAE’s limited latent coverage?
> The main goal of our paper was to evaluate whether ECo-MoE can increase the evolvability of complex freeform robots relative to the state-of-the-art, which used a monolithic universal controller. Evo by Demo was intended as an auxiliary, exploratory extension of ECo-MoE, something that is now possible given our modular approach.
> The quality of steering is fundamentally constrained by VAE, which we adopted from Li et al. (2025), and which does not generalize to out of domain data. We experimented with encoding and decoding a manually designed biped model in the material and had a lossy result, as shown in Fig. S9. For other simpler morphologies such as snake-like form, since they automatically emerge in the flat ground with trivial movement fitness, we do not perform steering experiments for them. We agree that more fine-grained steering mechanisms can possibly be achieved with more powerful generators with explicitly exposed control interfaces.
> ## How can a single predefined morphology produce 128 different voxelized representations in Fig. 3?
> In our revised manuscript we will clarify that this is due to properties of the adopted VAE embedding, which encodes each 64×64×64 voxel as a 10-channel one-hot vector: one channel for empty space, one for soft material, and eight for rigid-part identity. For a given robot design, we construct a segment-level adjacency graph over rigid parts and assign rigid-part labels using a seeded randomized graph-coloring procedure, ensuring that adjacent rigid parts receive different labels. For example, the simplest Predefined Design 2 in Fig. 3 of our original submission contains five rigid parts: one torso and four legs. The torso can take any of the eight rigid-part labels, and each leg must differ from the torso label. Since the legs are not adjacent to one another, each can independently choose any of the remaining seven labels, yielding 8×7^4=19,208 valid encodings of the same morphology. We sampled 128 such encodings per design using different random seeds, yielding the 128 latent vectors shown in Fig. 3.
> ## Why is MoE-based modularization preferable to hypernetwork-based alternatives?
> One advantage of MoE-based modularization in our setting is that it directly enables Evo by Demo: a frozen pretrained expert can be inserted into one slot of the mixture while the remaining experts and gate continue to train, with no architectural modification required. A hypernetwork entangles all morphology-control knowledge across its weights, offering no natural mechanism to inject or preserve an external policy without retraining the entire system. Additionally, hypernetwork training is known to be unstable which is a challenge acknowledged by Xiong et al. (2023), this could be further compounded in our co-evolutionary setting, where the morphology distribution shifts every generation. We will expand this conceptual comparison in the Related Work section of our revised manuscript.
> ## References
> - Li, M., Kong, L., and Kriegman, S. Generating freeform endoskeletal robots. In Proceedings of the International Conference on Learning Representations (ICLR), 2025.
> - Xiong, Z., Beck, J., and Whiteson, S. Universal morphology control via contextual modulation. In Proceedings of the International Conference on Machine Learning (ICML), pp. 38286–38300, 2023.

---

> > ### Author Rebuttal · Reviewer_ozBC · 2026-04-04
> >
> > I appreciate the authors' detailed reply and am happy to keep my score.

---

### Official Review · Reviewer_t7XA · 2026-03-07

**Soundness:** 3
**Presentation:** 4
**Significance:** 2
**Originality:** 3
**Overall Recommendation:** 4
**Confidence:** 3

**Summary:**

This work introduces an Embodiment-Conditioned Mixture of Experts (ECo-MoE) framework to jointly co-optimize the physical morphologies and neural controllers of evolving simulated robots. The main approach utilizes the continuous latent design vector of each robot as a gating signal to dynamically route and combine the action outputs of multiple neural control experts. Furthermore, the authors propose an "Evo by Demo" mechanism that explicitly steers the evolutionary search toward predefined structural traits by integrating a frozen, morphology-specific pre-trained expert into the routing network.

**Compliance With Llm Reviewing Policy:**

Affirmed.

**Final Justification:**

The rebuttal addressed all my concerns.

**Key Questions For Authors:**

1. **Unbiased PCA projection (Relates to W1):**
As stated in Lines 188-191, the latent dimensions are pre-filtered based on their correlation with fitness before PCA. This data leakage mathematically guarantees that high-performing policies traverse further along PC1. Can you provide a genuinely unbiased PCA projection (e.g., computed across the full 512-D space without any fitness-based filtering) to support the claim that ECo-MoE explores a broader region of the latent space (Figure 2E)?

2. **Resolving the methodological paradox of the "Evo by Demo" latent prior:**
Given the authors' explicit admission that the VAE latent mapping is unreliable and lacks stable correspondences (Lines 399-402), how does the predefined prior (Eq. 9) achieve genuine morphological steering rather than simply restricting the search to an arbitrary bounding box? To rule out the possibility that the CMA-ES is merely sampling tightly clustered clones of the predefined anchor, please provide quantitative morphological diversity or variance metrics for the final "co-steering" populations (Fig. 5M-O).

3. **Physical intuition of the routing diversity loss ($L_{div}$):**
To prevent expert collapse, the authors propose a routing diversity regularization loss ($L_{div}$, Equation 5). While I agree this is an effective practical engineering solution, its physical intuition merits further discussion. The authors define this loss to explicitly penalize similar routing weights for morphologically distant latents (measured by $L_2$ distance in the VAE space). However, in physical locomotion, it is entirely possible for structurally different morphologies (e.g., robots with different overall scales or geometries that share similar kinematic lever principles) to optimally utilize identical low-level sensorimotor strategies. Without this spatial penalty, does the system inevitably suffer from single-expert collapse, or do kinematically similar but morphologically distinct robots organically route to the same expert?

**Limitations:**

Yes.

**Strengths And Weaknesses:**

**Strengths:**
1. **Cross-Domain Formulation:**
By transplanting the Mixture-of-Experts(MoE) paradigm into the continuous morphological latent space of evolving physical agents. Using a frozen pre-trained Variational Autoencoder for morphology search is an elegant system-level metaphor.
2. **Robust Complex System Integration:**
Stably nesting a derivative-free evolutionary outer loop (CMA-ES) with a gradient-based reinforcement learning inner loop (PPO) within a high-dimensional voxel simulator is a non-trivial engineering achievement. Furthermore, as you correctly noted, the explicit incorporation of the routing diversity loss ($L_{div}$) is an effective engineering choice to mitigate the expert-collapse issue during this highly unstable co-optimization process.


**Weaknesses:**
1. **Statistical confounding in the exploration analysis:**
The methodology used to visualize and support the authors' claim that ECo-MoE "explores a broader region in the projected latent space" (Lines 206-207, Figure 2) suffers from data leakage. By explicitly stating that they pre-select the "100 latent dimensions most correlated with fitness" (Lines 184-185) before performing dimensionality reduction, the analysis introduces circular reasoning. This mathematically guarantees that higher-performing runs will inherently appear to have larger spatial displacements (as claimed in Figure 2D and 2E), rendering the exploration claims drawn from this PCA projection statistically invalid.

2. **Functional trade-offs in morphological steering:**
The "Evo by Demo" framework demonstrates that evolutionary search can be coerced into recovering predefined structural traits. However, the evaluation focuses exclusively on a reductive morphological similarity metric while fundamentally altering the evolutionary objective to heavily favor the pretrained expert. By omitting the raw, unaugmented task performance, the paper fails to assess whether forcing these structural priors severely compromises the robots' actual physical locomotion capabilities.

3. **Misalignment between physical intuition and algorithmic constraints:**
To prevent expert collapse, the framework employs a spatial diversity loss that forces morphologically distant robots (in the VAE latent space) to use different control experts. This constraint contradicts the physical reality of locomotion, as structurally diverse morphologies can frequently and optimally share identical kinematic strategies. Combined with the authors' admission that the VAE struggles to learn stable morphological correspondences, this penalty risks artificially preventing natural cross-embodiment generalization, making it ambiguous whether the observed modularity is physically organic or mathematically forced.

---

> ### Author Rebuttal · Authors · 2026-03-31
>
> We appreciate the reviewer’s thoughtful critique and their detailed questions regarding our analysis and interpretation of our results. Below we describe the new experiments and analysis we conducted to address the reviewer’s critique. We believe that the paper has been significantly improved as a result of this additional effort.
> ## Does ECo-MoE still appear to explore a broader region of latent space when PCA is computed without fitness-based filtering?
> We thank the reviewer for pointing out this discrepancy in our analysis. This was an oversight on our part and we completely agree that selecting latent dimensions by their correlation with fitness before performing PCA rendered the visualization somewhat uninformative. To address this, we redid this analysis using PCA over the full 512-D latent space. We provide this corrected analysis for all three tasks (Figs. S2-S4) in our anonymous supplementary material here: https://inspiring-cendol-ab20d0.netlify.app. We show these new results alongside the original biased projection method for reference but will remove it from our revised manuscript. The corrected analysis shows that for the Flat Ground task, ECo-MoE and the Baseline have similar task performance, net displacement, and traversed area under both projection methods. For the Upright Locomotion task, the original projection suggests larger displacement and area for ECo-MoE, but under PCA over all latent dimensions, ECo-MoE and the Baseline show broadly similar displacement and area. For the Potholes task, although ECo-MoE achieves better task performance, both methods again exhibit similar displacement and traversed area under both projections. Overall, these results suggest that the advantage of ECo-MoE does not primarily come from exploring farther in latent space, but rather from more effectively controlling and exploiting the designs encountered during evolution. We once again thank the reviewer for helping us improve our analysis and we will revise the discussion in the paper accordingly.
> ## Does Evo by Demo steer morphology without sacrificing raw locomotion performance, and does it produce meaningful population-level variation rather than tightly clustered clones?
> We conducted additional analysis to address both points and report results in https://inspiring-cendol-ab20d0.netlify.app. In Fig. S7, we evaluate the final evolved populations at generation 60 using two complementary shape metrics: the effective limb count and the mass bias vector magnitude. The within-population standard deviations under design-only and co-steering are comparable to or larger than those of the unsteered baselines across both metrics and all three designs, ruling out the possibility that CMA-ES is merely sampling tightly clustered clones and confirming that the latent prior preserves meaningful phenotypic diversity. To verify statistical stability and address the concern about raw task performance, we conducted repeated trials on predefined Design 1 (Fig. S8): three seeds each for PretrainOnly, DesignOnly, and CoSteering, and five seeds each for Baseline and default ECo-MoE. Fig. S8(A, B) confirms that the morphology steering trends from Fig. 5 are robust across seeds. Fig. S8(C) reports the raw, unaugmented task fitness with 95% bootstrapped confidence intervals: co-steering and design-only both substantially outperform the Baseline and default ECo-MoE, while pretrain-only underperforms both. This demonstrates that Evo by Demo does not recover predefined traits at the expense of locomotion capability.
> ## What is the role of the routing diversity loss, and does ECo-MoE collapse to a single expert without it?
> We agree that morphologically distinct robots may still benefit from similar low-level control strategies, so morphology-dependent routing should not be interpreted as requiring fully isolated experts for different bodies. Our routing diversity loss is intended primarily as a practical stabilization mechanism to prevent expert collapse during joint co-optimization, rather than as a strict physical prior.
> To directly address this concern, we provide additional ablations and routing analyses here: https://inspiring-cendol-ab20d0.netlify.app. Without routing diversity regularization, ECo-MoE tends to collapse early to a single dominant expert used by nearly the entire population (Fig. S5 B). With the regularization enabled, expert usage remains distributed across multiple experts (Fig. S5 A). At the same time, routing is not one-to-one: robots are often controlled by mixtures of experts rather than completely isolated experts, which still allows partial sharing across embodiments.

---

> > ### Author Rebuttal · Reviewer_t7XA · 2026-04-02
> >
> > I thank the authors for their detailed clarifications, which resolve most of my concerns. Therefore, I will keep the positive rating of **(4: Weak Accept)**.

---

### Official Review · Reviewer_Aprb · 2026-03-13

**Soundness:** 3
**Presentation:** 3
**Significance:** 3
**Originality:** 3
**Overall Recommendation:** 5
**Confidence:** 4

**Summary:**

This paper proposes Embodiment-Conditioned Mixture of Experts (ECo-MoE), a controller architecture for evolutionary robotics in which a shared set of expert policies is combined through a gating  network conditioned on a robot’s morphology latent vector. Robot morphologies are sampled from a latent design distribution and evolved using CMA-ES, while the controller parameters are optimized with PPO.

In contrast to previous work the paper builds on, the approach aims to address the challenging trade-off between training a separate controller per robot morphology and using a single universal controller. By conditioning expert routing on morphology, the method allows specialization while still sharing knowledge  across designs. Experiments on locomotion tasks show that the proposed controller improves evolvability and final fitness in environments that require more diverse morphologies. The paper also introduces a interesting extension called "evo by demo" mechanism in which pretrained experts can guide evolution toward particular morphological traits.

**Compliance With Llm Reviewing Policy:**

Affirmed.

**Final Justification:**

The author addressed all my remaining concerns.

**Key Questions For Authors:**

- How frequently is each expert used during training and evaluation, and do experts specialize for different regions of the morphology latent space?
- Does the routing diversity regularization actually increase diversity in expert usage?
- Did you observe any cases of expert collapse or underutilization during training?
- How sensitive are the results to the number of experts in the mixture?

**Limitations:**

yes

**Strengths And Weaknesses:**

Strengths:

The paper is interesting and well written. The core idea of conditioning a mixture-of-experts controller on robot embodiment is also intuitive and well motivated. It provides a principled middle ground between morphology-specific controllers and universal policies, which is an important problem in evolutionary robotics. The integration of reinforcement learning and evolutionary search is clearly described and build  on prior work. The “evo by demo” mechanism is also an interesting extension that allows demonstration policies to bias morphology search toward desired structures. Empirically, the method appears to improve performance in tasks that require greater morphological diversity or environmental complexity.

Weaknesses:

For me, the main limitation of the paper is that the behavior of the mixture-of-experts architecture is not analyzed in detail. The authors introduce a routing diversity regularization term intended to encourage different morphologies to activate different experts, but the paper does not provide empirical evidence that this mechanism actually leads to expert specialization (as far as I could tell). For example, there are no analyses of gating distributions, expert utilization, or routing entropy, which would be helpful to add. Without such evidence, it is difficult to verify whether the mixture-of-experts mechanism is functioning as intended or whether performance gains arise primarily from other factors.

Additionally, the paper could benefit from further ablation studies that would help clarify the role of key design choices. In particular, the effect of the routing diversity regularizer, the number of experts, and the conditioning strategy for the gating network are not explored. More minor, the evaluation is also limited to locomotion tasks, which makes it difficult to assess how well the approach generalizes to other forms of embodied behavior such as manipulation.

---

> ### Author Rebuttal · Authors · 2026-03-31
>
> We thank the reviewer for their thoughtful and encouraging feedback. We appreciate the questions regarding the internal behavior of ECo-MoE and the need for additional analysis. Below we address the reviewer’s comments to the best of our ability and describe the experiments we conducted to answer the reviewer’s questions. We feel that this exercise has strengthened our paper considerably.
> ## What was the effect of routing diversity regularization?
> To answer this question we conducted an ablation study in which routing diversity regularization was removed. The results of this experiment are included in the anonymous supplementary material (Fig. S5), which can be viewed at: https://inspiring-cendol-ab20d0.netlify.app. For the reviewer’s convenience, we briefly summarize the findings here. At any generation of evolution, we can analyze how routing weights were distributed on average for each robot in the population in terms of expert rank (from most heavily weighted to least) and across each robot in the population in terms of expert index. With routing diversity regularization enabled, expert usage remains distributed across multiple experts throughout evolution, allowing different robot designs to be controlled by different experts or mixtures of experts. Without this regularization, however, ECo-MoE quickly concentrates routing weight onto a single expert and eventually routes nearly the entire population through that expert, leading to expert collapse. These results indicate that routing diversity regularization is indeed important for preventing collapse and maintaining morphology-dependent specialization across the evolving population.
> ## How were experts utilized?
> We have analyzed the utilization of experts and provide additional figures illustrating expert usage in the anonymous supplementary material (Fig. S1), which can be found here: https://inspiring-cendol-ab20d0.netlify.app. In brief, we visualize the top 30 robots in generation 120 of evolution for Upright Locomotion, ranked by fitness, together with their routing weight distributions over the four experts. This analysis shows that, in later generations, different high-performing robot morphologies are routed to different mixtures of experts, demonstrating morphology-dependent routing behavior. We additionally tracked how expert usage evolves over generations using the same routing-distribution statistics described above. We found that, as evolution progresses, each robot tends to place more weight on a smaller subset of experts, indicating increasing specialization. At the same time, the most-used experts change over time, suggesting that ECo-MoE dynamically adapts expert usage as the population of robot designs evolves. We find this quite interesting and once again thank the reviewer for urging us to carry out this analysis.
> ## How sensitive are the results to the number of experts?
> To measure the sensitivity of our approach to the number of experts, we performed additional trials of ECo-MoE using half as many experts (two) and twice as many experts (eight). Consistent with the Baseline (which may be considered a single expert) and the main ECo-MoE trials (which have four experts), we adjusted the size of the experts based on their number so that all variants have a similar overall parameter budget. The parameter table (Table S1) and results (Fig. S6) can be seen here: https://inspiring-cendol-ab20d0.netlify.app. In summary, all ECo-MoE variants outperform the Baseline, suggesting that the benefit of embodiment-conditioned expert routing is broadly robust to the selected number of experts. With two experts, both experts are used cooperatively across the population, but the routing patterns appear less sensitive to morphological differences. With eight experts, only a subset of experts was consistently utilized with the remainder receiving near-zero routing weight throughout evolution. Under the current settings, four experts seem to provide the best balance among performance, specialization, and effective parameter usage.

---

> > ### Author Rebuttal · Reviewer_Aprb · 2026-04-03
> >
> > Thank you for addressing my question and for adding these additional experiments. Analysing the MoE workings in more detail significantly increased the impact of the paper for me; I will increase my score.

---

### Decision · Program_Chairs · 2026-04-30

**Decision:**

Accept (regular)

**Comment:**

The paper proposes ECo-MoE, a Mixture-of-Experts controller for co-optimizing robot morphology and control via evolutionary methods, together with a novel "Evo by Demo" paradigm that steers evolution using pre-trained expert controllers.

The reviewers appreciated the clarity of the paper's presentation, the novelty of applying MoE to the morphology-control co-optimization problem, and the creative Evo by Demo mechanism that naturally emerges from the MoE architecture. The rebuttal was exceptionally thorough and candid. In particular, the authors corrected their PCA analysis after Reviewer t7XA identified a methodological confound -- revising their own narrative from "explores farther" to "exploits more effectively" -- which all reviewers noted as a sign of scientific rigor. New ablations over the number of experts K, the routing diversity regularizer L_div, and multi-seed experiments with bootstrapped confidence intervals substantially strengthened the empirical contribution. Gating visualizations (Fig. S1) confirmed that the MoE routes morphologically distinct robots to different experts, validating the core hypothesis.

Remaining limitations are clearly scoped: the evaluation is restricted to locomotion tasks, no empirical hypernetwork comparison is provided (though a reasonable conceptual argument is given), and the VAE morphology encoder limits the range of achievable body plans. These are directions for future work rather than fundamental shortcomings.

The paper makes a technically sound, clearly written, and original contribution to the intersection of evolutionary robotics and modular neural architectures. The Evo by Demo capability is, to my knowledge, novel and opens an interesting research direction. I recommend acceptance.